# *Lactobacillus**johnsonii* L531 Protects against *Salmonella* Infantis-Induced Intestinal Damage by Regulating the NOD Activation, Endoplasmic Reticulum Stress, and Autophagy

**DOI:** 10.3390/ijms231810395

**Published:** 2022-09-08

**Authors:** Lan Yang, Jiu-Feng Wang, Ning Liu, Xue Wang, Jing Wang, Guang-Hui Yang, Gui-Yan Yang, Yao-Hong Zhu

**Affiliations:** 1College of Veterinary Medicine, China Agricultural University, Beijing 100193, China; 2Department of Pathology and Laboratory Medicine, University of California at Davis, Sacramento, CA 95616, USA

**Keywords:** *Salmonella*, probiotic, *Lactobacillus johnsonii*, NOD, autophagy, endoplasmic reticulum stress

## Abstract

*Salmonella enterica* serovar Infantis (*S.* Infantis) is an intracellular bacterial pathogen. It is prevalent but resistant to antibiotics. Therefore, the therapeutic effect of antibiotics on *Salmonella* infection is limited. In this study, we used the piglet diarrhea model and the Caco2 cell model to explore the mechanism of probiotic *Lactobacillus johnsonii* L531 (*L. johnsonii* L531) against *S.* Infantis infection. *L. johnsonii* L531 attenuated *S.* Infantis-induced intestinal structural and cellular ultrastructural damage. The expression of NOD pathway-related proteins (NOD1/2, RIP2), autophagy-related key proteins (ATG16L1, IRGM), and endoplasmic reticulum (ER) stress markers (GRP78, IRE1) were increased after *S.* Infantis infection. Notably, *L. johnsonii* L531 pretreatment not only inhibited the activation of the above signaling pathways but also played an anti-*S.* Infantis infection role in accelerating autophagic degradation. However, RIP2 knockdown did not interfere with ER stress and the activation of autophagy induced by *S.* Infantis in Caco2 cells. Our data suggest that *L. johnsonii* L531 pretreatment alleviates the intestinal damage caused by *S.* Infantis by inhibiting NOD activation and regulating ER stress, as well as promoting autophagic degradation.

## 1. Introduction

*Salmonella enterica* serovar Infantis (*S.* Infantis) is a pathogenetic bacteria not only in animals but also in humans. It causes fever, intestinal inflammation, and diarrhea [1]. Due to multi-drug resistance and limitations in the therapeutic effect of using antibiotics [2], probiotics in the prevention of *Salmonella* infection have been an alternative. Our previous studies have found that *Lactobacillus johnsonii* L531 (*L. johnsonii* L531) is effective in preventing *S.* Infantis-induced diarrhea and intestinal inflammation in a piglet model [3,4,5]. The anti-inflammatory effect of *Lactobacillus johnsonii* was proposed to be associated with endoplasmic reticulum (ER) stress [5,6]. However, the detailed mechanism of how *L. johnsonii* L531 regulates ER stress and related signaling pathways has not been well studied.

The ER is an important location for protein folding, signaling, and lipid biosynthesis. The unfolded protein response (UPR) is precise signaling evolved by ER to maintain a stable intracellular environment [7]. Bacterial infection can result in the accumulation of unfolded or misfolded proteins within ER. When ER stress occurs, GRP78 dissociates from transmembrane receptors that make up UPR, activating the nuclear transcription factor-κB (NF-κB) pathway and IL-6 production [8,9]. It may be the leading cause of inflammation and intestinal damage [10]. Previous studies revealed that probiotics attenuated intestinal inflammation by reducing the activities of ER stress markers as well as downregulating UPR [6,11,12]. Recent findings indicate that ER stress-induced proinflammatory responses depend on NOD-like receptors (NLRs; NOD1/NOD2) [13].

NOD1 and NOD2 sense *Salmonella* invasion and activate NF-κB-dependent inflammatory response via receptor-interacting protein 2 (RIP2) [14,15,16]. Consequently, autophagy activation was induced by pathogen-associated molecular pattern molecules (PAMPs) in the cytoplasm. Autophagy, a lysosomal degradation process, plays a crucial role in maintaining protein metabolism and the stability of the cellular environment. Although autophagy is an important host defense process, bacteria have evolved strategies to escape it. *Legionella pneumophila* inhibits the formation of autophagosomes by cutting the lipid chain of LC3-PE, thereby achieving the goal of survival by regulating autophagy [17]. *Shigella flexneri* delivers virulence effectors IcsB into host cells, which alters the host membrane of the fatty acylation landscape to get rid of host autophagy [18]. Furthermore, inhibition of autophagy significantly reduces the over-replication of cytoplasmic *S.* Infantis [19]. NOD1/2 recruits ATG16L1 and induces autophagy to the site of intracellular bacteria invasion to remove pathogens [20,21]. However, there is no consensus on whether this process is through RIP2 or direct binding to ATG16L1.

In the current study, we used *S.* Infantis infection models in piglets and Caco2 cells to elucidate the effects of *L. johnsonii* L531 on regulating the NOD pathway, ER stress, and autophagy. Additionally, the role of RIP2 in ER stress and autophagy during *S.* Infantis infection in Caco2 cells was also covered.

## 2. Results

### 2.1. L. johnsonii L531 Is Protective in the Prevention of S. Infantis-Induced Intestinal Inflammation and Damage to the Small Intestine

Previous findings have shown that *Lactobacillus johnsonii* L531 effectively alleviates *Salmonella*-induced diarrhea and small intestinal inflammation [3,4,5,22]. Our present study also demonstrated that the SI group had more extensive tissue destruction, including intestinal gland atrophy, epithelium loss on the mucosal surface, and increased inflammatory cells in the lamina propria. Pretreatment with *L. johnsonii* L531 attenuates *S.* Infantis–induced tissue damage in the small intestine. The small intestinal histology in this study is shown in Appendix A.

### 2.2. Pretreatment with L. johnsonii L531 Suppresses S. Infantis-Induced Activation of the NOD Pathway in the Small Intestine

Activation of NOD1 and NOD2 by peptidoglycans induces and initiates autophagy, which is important in ER stress-induced inflammation. Immunofluorescence results showed that *S.* Infantis infection increased the number of NOD1-positive cells in the SI group but not in the LS group in the jejunum (Figure 1A). Western blotting analysis also showed that the protein levels of NOD1, NOD2, and its primary adapter protein RIP2 in the jejunum were higher in the SI group than in the CN group. However, NOD1/2 protein expression in the LS group showed no significant difference in comparison with the CN group (Figure 1B). The protein levels of NOD2, NOD1, and RIP2 in the ileum were not significantly changed (Figure 1B).

### 2.3. Pretreatment with L. johnsonii L531 Attenuates S. Infantis-Induced Autophagy in the Small Intestine

ATG16L1, an essential autophagy protein, plays a key role in bacterial clearance. Immunofluorescence results showed that *S.* Infantis infection increased the ATG16L1-positive staining area in jejunal mucosa, including lamina propria (Figure 2A). Consistently, Western blot results showed that the levels of autophagy-related proteins (ATG16L1 and IRGM) were significantly increased in the jejunum and ileum by *S.* Infantis. Pretreatment with *L. johnsonii* L531 could prevent the increased expression of ATG16L1 and IRGM induced by *S.* Infantis (Figure 2B).

In addition, *S.* Infantis upregulated the mRNA expression of *Il*6 in the small intestine (Figure 2C). As *Il*6 is proinflammatory, our data indicate that *L. johnsonii* L531 can attenuate intestinal inflammation and autophagy caused by *S.* Infantis in a piglet model.

### 2.4. L. johnsonii L531 Represses S. Infantis-Induced NOD Activation and Modulates Autophagy in Caco2 Cells

We further investigated the protein expression of NOD1, NOD2, downstream RIP2, as well as autophagy-related proteins in Caco2 cells after *S.* Infantis infection. Compared with the control group, *S.* Infantis elevated the expression of NOD1, NOD2, and RIP2 at 5 h post-infection. Pretreatment with *L. johnsonii* L531 for 3 h before the *S.* Infantis challenge effectively reversed these phenomena (Figure 3A).

Compared with the control group, *S*. Infantis also induced an increase in ATG16L1, IRGM, and LC3-II protein levels and a decrease in P62 level in Caco2 cells, indicating that *S.* Infantis infection activated autophagy in epithelial cells. Notably, *L. johnsonii* L531 pretreatment downregulated the activation of autophagy induced by *S*. Infantis (Figure 3B). Immunofluorescence revealed an increase in co-localization of NOD1 and ATG16L1 in Caco2 cells by *S.* Infantis invasion (Figure 3C). These data suggest that an interaction between NOD1 and ATG16L1 may facilitate the RIP2 non-dependent autophagy triggered by *S.* Infantis.

### 2.5. L. johnsonii L531 Enhances the Resistance of Caco2 Cells to S. Infantis by Promoting Autophagy Degradation

To further investigate the effect of *L. johnsonii* L531 on the autophagy of intestinal epithelial cells induced by *S.* Infantis, Caco2 cells were treated with CQ, an autophagy degradation inhibitor, for 2 h to inhibit the autophagy degradation. *S*. Infantis infection significantly elevated the expression of LC3-II in Caco2 cells (Figure 4A). As expected, CQ treatment increased LC3 accumulation in all cell groups. There was no difference in LC3-II protein expression between control cells without CQ treatment and *L. johnsonii* L531 pretreated cells, suggesting that *L. johnsonii* L531 might not impact autophagosome synthesis. Notably, in the presence of CQ, LC3-II protein level was significantly lower in cells treated with *L. johnsonii* L531 than in the untreated group (Figure 4B). Together with these data, *L. johnsonii* L531 may facilitate autophagy degradation of Caco2 cells in response to *S.* Infantis infection.

### 2.6. L. johnsonii L531 Alleviates Cell Damage by Regulating ER Stress in Response to S. Infantis

To investigate the effect of *L. johnsonii* L531 against *S.* Infantis, the morphology of Caco2 cells was observed by scanning electron microscopy (SEM). In comparison to the control group, *S.* Infantis challenge resulted in shedding, rarefaction of microvilli on the cell surface, and ruffling in some areas of the cell membrane. Besides, part of the *S.* Infantis perforated in the cell membrane and extruded towards the apical side, disrupting the cell membrane integrity. Pretreatment of *L. johnsonii* L531 effectively alleviated the ultrastructural damage to the cell surface caused by *S.* Infantis (Figure 5A). Transmission electron microscopy (TEM) revealed that in *S.* Infantis-infected Caco2 cells, the lumens of ER were severely expanded, vacuoles were formed, as well as more autophagolysosomes were observed in the cytoplasm. *L. johnsonii* L531-treated cells did not have these ultrastructural damages (Figure 5B).

Remarkably, *S.* Infantis, ER stress activator thapsigargin (TG), and tunicamycin (TM) increased the protein expression of GRP78, IRE1, EIF2S1 as well as the gene expression of *Il*6 and *DDIT3* in Caco2 cells compared with the control (Figure 5C,D). However, this increase was inhibited by *L. johnsonii* L531 (Figure 5C,D).

### 2.7. RIP2 Inhibition Did Not Alter S. Infantis-Induced Autophagy and ER Stress in Caco2 Cells

NOD proteins mediate LC3-II recruitment to vesicles that contain internalized bacteria to initiate autophagy. Whether this process is through RIP2 signaling or direct binding between the NLRs and ATG16L1 is controversial [14,23,24]. We hypothesize that NOD1 may directly interact with ATG16L1, bypassing RIP2 during *S.* Infantis infection to initiate autophagy. To investigate whether the protein adaptor RIP2 is required for NOD1-initiated autophagy after *S.* Infantis infection, we applied siRNA technology to knock down RIP2 protein in Caco2 cells. Western blotting assay showed that siRIP2 effectively suppressed the expression of RIP2 in this study (Figure 6A). However, inhibition of RIP2 expression did not affect the expression of NOD1 and NOD2 proteins during *S.* Infantis infection (Figure 6B). Additionally, there were no differences in the protein expression of P62, IRGM, ATG16L1, LC3-II, GRP78, and IRE1 in siRIP2-treated *S.* Infantis-challenged Caco2 cells compared with untreated groups (Figure 6C,D).

## 3. Discussion

In this study, we demonstrate that *L. johnsonii* L531 pretreatment effectively ameliorates *S.* Infantis-induced NOD1/2 activation, ER stress, and autophagosome degradation, thereby contributing to alleviating intestinal tissue and cell damage. However, RIP2 is not dispensable during ER stress and autophagy induced by *S.* Infantis (Figure 7).

*S.* Infantis infection in newly weaned piglets causes gut microbiota dysbiosis and results in enteritis, diarrhea, fever, and poor growth performance [1,23,24,25]. *Lactobacillus* has been shown protective against *Salmonella* infection [3,4,5,18,23]. In the piglet model, *L. johnsonii* L531 could increase SCFAs levels and downregulate the NF-κB-SQSTM1 mitophagy signaling pathway, thus, alleviating intestinal inflammation and limiting *S.* Infantis dissemination [3,4].

NLRs are important in innate immune response and inflammation development. NLRP3 and NLRC4 could sense *Salmonella* invasion and subsequently trigger cell death pathways to eliminate pathogens [26]. NOD1 and NOD2 promote intestinal inflammation by RIP2-mediated activation of NF-κB [4,22,27]. Furthermore, NOD1 and NOD2 are vital for the formation of autophagosomes. Autophagy can transport intracellular bacteria to lysosomes, where they can be neutralized [28]. A recent study has found that autophagy plays a salient role in *Salmonella* contagion. It is beneficial to moderate the activation of autophagy which inhibits the proliferation of *Salmonella* [29]. It is noted that *Salmonella*, as a vacuolar bacterium, can commandeer autophagy by inhibiting its delivery to the lysosomal compartment [30]. In this situation, intracellular pathogens turn autophagosomes into protective shelters. IRGM is vital in the autophagy pathway and is essential for bacterial clearance [31]. Our data showed *S.* Infantis infection significantly increased the expression of ATG16L1 and IRGM in the small intestine and Caco2 cells. Meanwhile, *L. johnsonii* L531 pretreatment attenuates *S.* Infantis-induced autophagy. *S.* Infantis infection had a similar effect to CQ, as they both led to the accumulation of LC3 protein in Caco2 cells. Therefore, we conclude *S.* Infantis may escape host immune responses by inhibiting autophagy degradation. By contrast, *L. johnsonii* L531 mitigated the accumulation of LC3 protein induced by CQ. These results indicate that *L. johnsonii* L531 regulates *S.* Infantis-induced autophagic activation in Caco2 cells by accelerating autophagic degradation.

NOD1 and NOD2 are also involved in ER stress-induced inflammation. The NF-κB pathway can be activated through IRE1, initiating an inflammatory response. Probiotics such as Bifidobacterium species have a protective effect on ER stress in Caco2 monolayers [32]. Beyond that, *L. johnsonii* L531 downregulated the mRNA expression of inflammatory factor IL-6 during *S.* Infantis infection, contributing to ER homeostasis [8]. Our results revealed that *S*. Infantis infection simultaneously activated the NOD pathway and ER stress in vivo and in vitro. It suggests that NOD1/2 and ER stress might act synergistically in protecting the small intestine from *S*. Infantis infection.

In addition to protein synthesis, ER is also the primary source of autophagy separation membranes [33,34]. *L. johnsonii* L531 pretreatment not only mitigated the release of *Il*6 but also prevented the overexpression of GRP78, IRE1, EIF2S1, and *DDIT3*. As confirmed by the Caco2 cell infection model, *L. johnsonii* L531 inhibits *S.* Infantis-induced ER stress and protects the ultrastructure of ER. *Salmonella* can use autophagy to promote its immune escape and reproduction [35,36]. Our results showed that *S.* Infantis could inhibit autophagy degradation. Considering the consequential damage to ER, we speculate that *S.* Infantis may abduct autophagosomes to achieve immune escape by acting on the ER. *L. johnsonii* L531 promotes autophagosome degradation and restores autophagic flux. Therefore, our results demonstrate that *L. johnsonii* L531 inhibits ER stress, thereby alleviating cell damage caused by *S.* Infantis.

In the presence of intracellular bacteria, NOD1 induces activation of the NF-κB signaling pathway through the adaptor RIP2 [27]. Meanwhile, NOD1 initiates autophagy by recruiting LC3 to *S.* Infantis-containing vacuole. Our data showed *S.* Infantis infection of Caco2 cells with or without siRIP2 could cause a significant elevation of autophagy proteins and ER stress-related proteins.

## 4. Materials and Methods

### 4.1. Ethics Statement

All the experimental animals in this study were treated in strict accordance with the Guidelines for Laboratory Animal Use and Care from the Chinese Center for Disease Control and Prevention and the Rules for Medical Laboratory Animals (1998) from the Chinese Ministry of Health, under protocol CAU20161016-1, which was approved by the Animal Ethics Committee of the China Agricultural University. All animals were euthanized under pentobarbital sodium anesthesia and every effort was made to alleviate the pain. Our experimental design met the agenda of OneHealth and the principles of the 3Rs (https://www.nc3rs.org.uk/), (accessed on 15 August 2022).

### 4.2. Bacterial Strains and Preparation

*Lactobacillus johnsonii* L531 was isolated from the colon contents of healthy neonatal weaned piglets [3,4]. *Salmonella enterica* serovar Infantis strain CAU1508 was isolated from the intestinal contents in diarrhea piglets [3,4].

### 4.3. Animals and Specimens

Jejunal and ileal tissues were derived from piglets, as previously described [3]. Briefly, piglets were divided into 3 groups on day 0 (6 animals per group): control group (CN), *S*. Infantis (SI) group (untreated group), and *L. johnsonii* + *S.* Infantis (LS) group. SI and LS piglets were challenged with 10 mL of *S.* Infantis (1.0 × 10^11^ CFU/mL) on day 8. LS piglets were administered 10 mL of *L. johnsonii* L531 (1.0 × 10^10^ CFU/mL) once daily from day 1 to 7. Animal necropsy was performed on day 18.

### 4.4. Regents and Antibodies

The siRNA for RIP2 was obtained from Shanghai GenePharma Co., Ltd. (Shanghai, China). Lipofectamine™ RNAiMAX transfection reagent (13778075) was purchased from ThermoFisher Scientific (Rockford, USA). The following primary antibodies were used: mouse NOD1 monoclonal antibody (sc-398696, Santa Cruz Shanghai, China), rabbit NOD2 polyclonal antibody (bs-7084R, Bioss, Beijing, China), rabbit RIP2 polyclonal antibody (bs-3546R, Bioss, Beijing, China), rabbit IRGM polyclonal antibody (ab69494, Abcam, Boston, MA, USA), rabbit IRE1 polyclonal antibody (ab37073, Abcam, Boston, MA, USA), rabbit p62/SQSTM1 polyclonal antibody (18420-1-AP, Proteintech, Rosemont, IL, USA), mouse GAPDH monoclonal antibody (60004-1-lg, Proteintech, Rosemont, IL, USA), mouse Beta ACTIN monoclonal antibody (60008-1-lg, Proteintech, Rosemont, IL, USA), rabbit ATG16L1 monoclonal antibody (#8089, Cell Signaling Technology Shanghai, China), rabbit GRP78 polyclonal antibody (#3177, Cell Signaling Technology Shanghai, China), rabbit EIF2S1 monoclonal antibody (#5324, Cell Signaling Technology Shanghai, China), and rabbit LC3A/B polyclonal antibody (#4108, Cell Signaling Technology Shanghai, China). The secondary antibodies were HRP-conjugated Affinipure Goat Anti-Rabbit IgG (H + L) (SA00001-2, Proteintech, Rosemont, IL, USA) and HRP-conjugated Affinipure Goat Anti-Mouse IgG (H + L) (HS201-01, Proteintech, Rosemont, IL, USA).

### 4.5. In Vitro Caco2 Cell Infection Model

Caco2 cells were cultured in DMEM/High Glucose supplemented with 10% fetal bovine serum (FBS) and 1% penicillin-streptomycin at 37 °C, 5% CO_2_, 95% air, and 95% relative humidity. Cells were seeded in six-well cultured plates (1 × 10^6^ cells per well) under four circumstances: (i) medium individually; (ii) *S*. Infantis individually at a multiplicity of infection (MOI) of 20; (iii) cultivation with *L. johnsonii* L531 at MOI of 50, or (iv) pre-cultivation with *L. johnsonii* L531 at MOI of 50 for 3 h before exposure to *S.* Infantis. Pre-incubation with *L. johnsonii* L531, the cells were washed 3 times after 3 h with phosphate-buffered saline (PBS) and immediately challenged with *S.* Infantis at MOI of 20. At 30 min after *S.* Infantis stimulation, all 4 groups were rinsed 3 times with PBS to eliminate non-invaded *S*. Infantis. Then, to kill any remaining extracellular *S*. Infantis, we added 100 μg/mL gentamicin to cultivate cells for another 2 h. *L. johnsonii* L531 was not contained in the medium during the infection. The experiments were repeated at least three times.

### 4.6. RNA Interference

To verify whether the NOD-mediated autophagy is induced in response to their specific ligand RIP2, we performed gene silencing to knock down RIP2. Caco2 cells were seeded in 24-well plates. According to the manufacturer’s instructions, when the cells grew to 60–80% confluence, RIP2-siRNA (si-RIP2, sense 5′-GGGAAGUGUUAUCCAGAATT-3′; antisense 5′-UUUCUGGAUAACACUUCCCTT-3′), diluted with RNAiMAX transfection reagent. Caco2 cells were transfected under the same condition as described above. After 5 h, cells in each well were further cultured for 36 h in 2 mL of 10% FBS.

### 4.7. Western Blotting

Proteins were extracted from tissues (jejunum and ileum) or Caco2 cells using the RIPA buffer (Solarbio, Beijing, China) and 1 mM PMSF (Solarbio, Beijing, China). BCA Protein Assay kit (23227, ThermoFisher Scientific, Waltham, MA, USA) was used to quantify the protein concentration. Proteins were divided into 10% or 12% SDS-polyacrylamide gel electrophoresis and transferred to the polyvinylidene difluoride membranes (Roche, Basel, Swiss). Subsequently, proteins were blocked with 5% skim milk at 37 °C for 1 h and then were incubated with the following primary antibodies overnight at 4 °C: NOD1 (1:500), NOD2 (1:1000), RIP2 (1:1000), ATG16L1 (1:1000), IRGM (1:1000), P62 (1:1000), LC3A/B (1:1000), GRP78 (1:1000), IRE1 (1:1000) and EIF2S1 (1:1000). Secondary antibodies (1:5000) with corresponding species source were used to incubate at 37 °C for 1 h. The images were visualized by an ECL detection system (Tanon 6200 chemiluminescence imaging workstation, Tanon Science & Technology Co., Ltd. Shanghai, China).

### 4.8. Quantitative Real-Time PCR

Total RNA from jejunal and ileal tissues, as well as Caco2 cells, was extracted using EASYspin plus RNA extraction kit (Aidlab Biotechnologies, Beijing, China). cDNA was generated using the PrimeScriptTM RT Reagent Kit (RR047A, TaKaRa, Dalian, China). The mRNA expression levels were measured using an SYBR Green PCR Master Mix (LS2062, Promega, Madison, WI, USA). Target gene expression was normalized to reference gene *Gapdh*. Primer sequences for in vivo and in vitro samples are listed in Table 1 and Table 2, respectively. Caco2 cell samples were collected at 5 h post *Salmonella* infection. Primers were synthesized by Tsingke Biotechnology (Beijing, China).

### 4.9. Scanning Electron Microscopy and Transmission Electron Microscopy

After 5 h of *S*. Infantis challenge, Caco2 cells were harvested and fixed in 3% glutaraldehyde. The samples were processed as previously described [3].

### 4.10. Immunofluorescence

Jejunal and ileal tissues were fixed in 4% formaldehyde for 24 h and implanted in paraffin blocks. Caco2 cells were fixed with 4% paraformaldehyde for 7 min, later cultivated with 1% Triton-X 100 (T8787, Sigma-Aldrich, St. Louis, USA) for 15 min. Bovine serum albumin (2%) was used to block tissue and cell specimens for 1 h at room temperature. Then tissue specimens were incubated with NOD1 antibody (1:100) and ATG16L1 antibody (1:50) overnight at 4 °C. Alexa Fluor 488 goat anti-rabbit or Alexa Fluor 555 donkey anti-rabbit was incubated with specimens for 1 h after washing 3 times with PBS at room temperature. DAPI was used to stain cell nuclei for 13 min at room temperature. Staining was visualized and photographed under a confocal microscope (Leica, Leica Microsystems, Germany). Cell samples were incubated with ATG16L1 antibody (1:100) overnight at 4 °C. After 3 times washed with PBS, cells were incubated with Alexa Fluor 488 goat anti-rabbit secondary antibodies at room temperature for 1 h. After washing 3 times with PBS, cell samples were incubated with NOD1 antibody (1:50) overnight at 4 °C, then incubated with secondary antibody (Alexa Fluor 555-labeled donkey anti-mouse) at room temperature for 1 h. Hoechst 33342 was used to stain live cell nuclei for 10 min at room temperature. Cell staining was visualized and photographed under a confocal laser scanning microscope (Nikon A1).

### 4.11. Statistical Analysis

GraphPad Prism 8 was used for statistical analysis and figure generation. Parallel experiments have been carried out at least three times with similar results. Data are expressed as means ± SEM and with one-way ANOVA (Tukey’s test) or the t-test with Bonferroni correction. *p* < 0.05 was statistically significant.

## 5. Conclusions

Our data reveal that *L. johnsonii* L531 restricts the overexpression of proteins involved in the NOD pathway, ER stress, and autophagosome degradation caused by *S.* Infantis infection. These effects of *L. johnsonii* L531 contribute to ameliorating *S.* Infantis-induced intestinal tissue and cell damage, indicating a protective role of *L. johnsonii* L531 in response to enteropathogenic bacteria. The limitation of the current study is that we did not explore which components of *S.* Infantis caused NOD activation and ER stress. The protective effectors from probiotic *L. johnsonii* L531 still need to be investigated in the future. To better develop the antibacterial therapy using probiotics, the NOD signaling pathway, autophagy, and ER stress crosstalk should be further studied and verified in different cells.

## Figures and Tables

**Figure 1 ijms-23-10395-f001:**
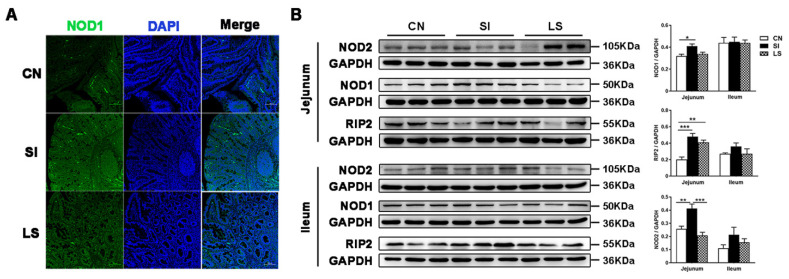
Oral administration of *L. johnsonii* L531 alleviates *S.* Infantis-induced activation of the NOD pathway. (**A**) Confocal micrograph illustrating the accumulation of NOD1 (Green) in the jejunum of pigs challenged with *S.* Infantis or *L. johnsonii* L531. Blue: DAPI. Scale bar, 100 μm. (**B**) Western blotting analysis for NOD2, NOD1, and RIP2 in the jejunum and ileum from piglets 10 days after *S.* Infantis challenge. The right panels show the protein quantitation using Image J software. Data are represented as mean ± SEM, *n* = 6; * *p* < 0.05, ** *p* < 0.01, *** *p* < 0.001.

**Figure 2 ijms-23-10395-f002:**
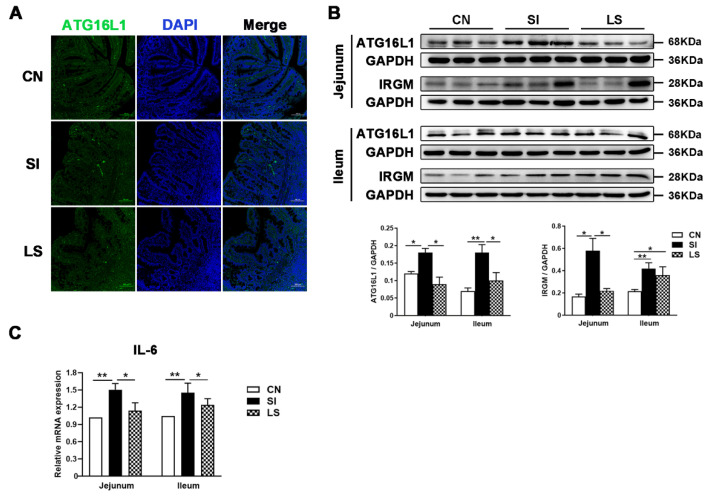
Oral *L. johnsonii* L531 attenuates the *S.* Infantis-induced jejunal and ileal autophagy. (**A**) Confocal micrograph illustrating the accumulation of ATG16L1 (Green) in the jejunum of pigs challenged with *S.* Infantis or *L. johnsonii* L531. Blue: DAPI. Scale bar, 100 μm. (**B**) Western blotting analysis for ATG16L1and IRGM in the jejunum and ileum from piglets 10 days after *S.* Infantis challenge. The right panels show the protein quantitation using Image J software. (**C**) Expression of *Il*6 mRNA in jejunal and ileal tissues from uninfected, *S.* Infantis-infected, and *L. johnsonii* L531 pretreated piglets on day 18. Data are represented as mean ± SEM, *n* = 6; * *p* < 0.05, ** *p* < 0.01.

**Figure 3 ijms-23-10395-f003:**
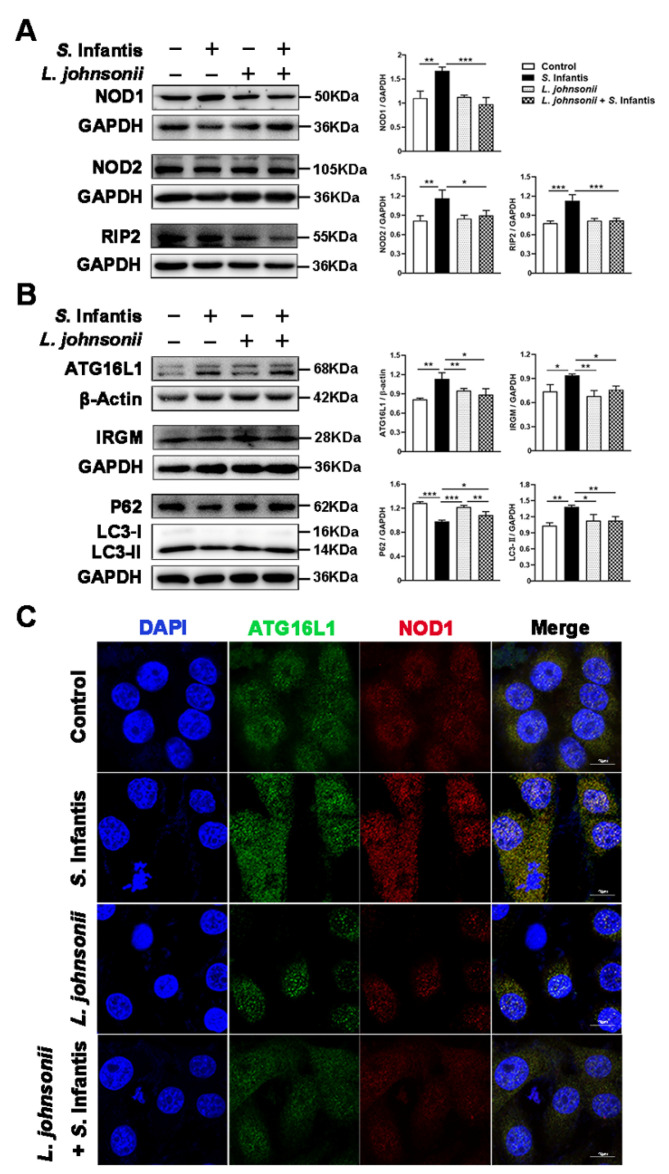
*L. johnsonii* L531 alleviates recruitment of ATG16L1 triggered autophagy by NOD1 after *S*. Infantis infection. (**A**,**B**) Western blotting analysis for NOD1, NOD2, RIP2, ATG16L1, IRGM, P62, and LC3 in Caco2 cells. The right panels show the protein quantitation using Image J software. (**C**) Confocal micrographs illustrating the accumulation of ATG16L1 (Green) and NOD1 (Red) in Caco2 cells treated with *S.* Infantis or *L. johnsonii* L531. Blue: DAPI. Scale bar, 10 μm. Data are represented as mean ± SEM, *n* = 3; * *p* < 0.05, ** *p* < 0.01, *** *p* < 0.001.

**Figure 4 ijms-23-10395-f004:**
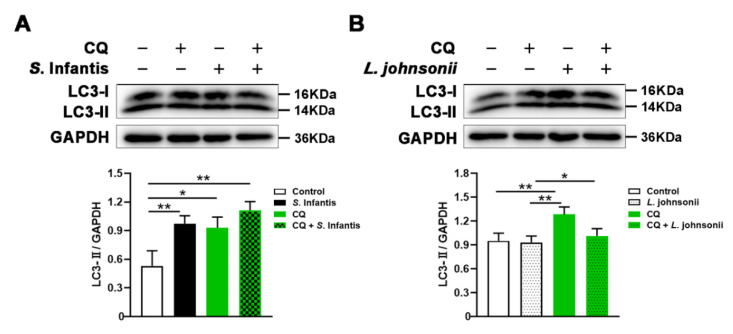
*S.* Infantis infection inhibits autophagy degradation of Caco2 cells, whereas *L. johnsonii* L531 accelerates this process. (**A**) Caco2 cells challenged with *S.* Infantis were pretreated with or without CQ. Western blotting analysis was conducted to detect the LC3 level. The lower panel shows the protein quantitation using Image J software. (**B**) Caco2 cells challenged with *L. johnsonii* L531 were pretreated with or without CQ. Western blotting analysis was conducted to detect the LC3 level. The lower panel shows the protein quantitation using Image J software. Data are represented as mean ± SEM, *n* = 3; * *p* < 0.05, ** *p* < 0.01, CQ, chloroquine.

**Figure 5 ijms-23-10395-f005:**
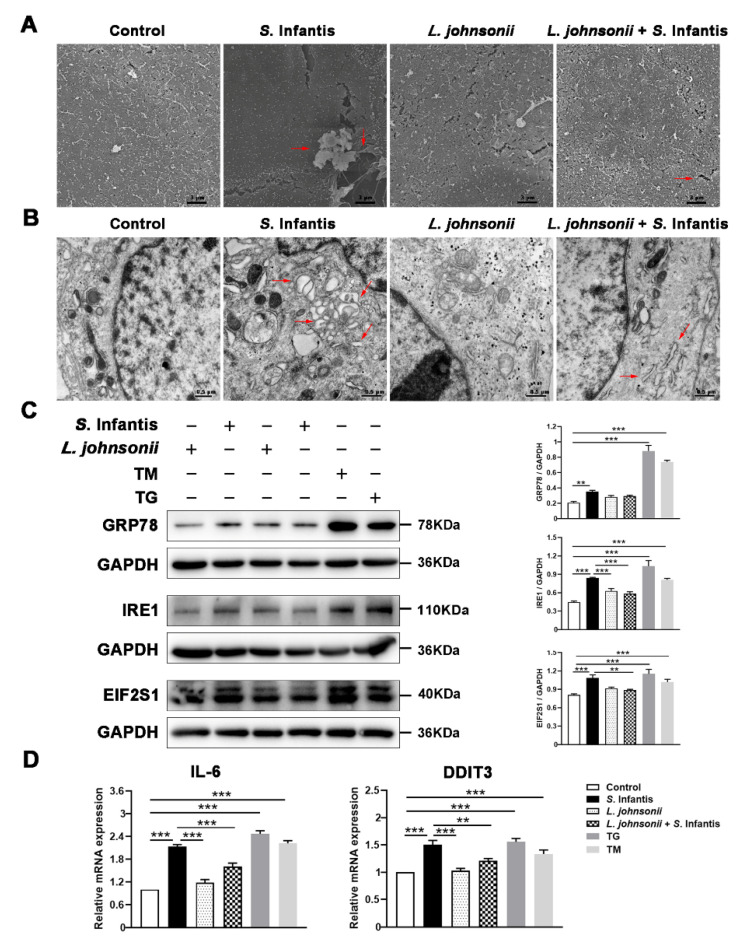
*L. johnsonii* L531 alleviates *S.* Infantis-induced ER stress and cellular ultrastructural damage in Caco2 cells. (**A**) Effect of *L. johnsonii* L531 on the structure of the ER in Caco2 cells after *S.* Infantis infection as observed using SEM. Red arrows indicate the damage caused by *S*. Infantis. (**B**) Effect of *L. johnsonii* L531 on the structure of apical surface in Caco2 cells after *S.* Infantis infection as observed using TEM. Red arrows indicate the damage caused by *S*. Infantis. (**C**) Western blotting analysis for GRP78, IRE1, and EIF2S1 in Caco2 cells. The right panels show the protein quantitation using Image J software. (**D**) Expression of *Il*6 and *DDIT3* mRNA in Caco2 cells. Data are represented as mean ± SEM, *n* = 3, ** *p* < 0.01, *** *p* < 0.001.

**Figure 6 ijms-23-10395-f006:**
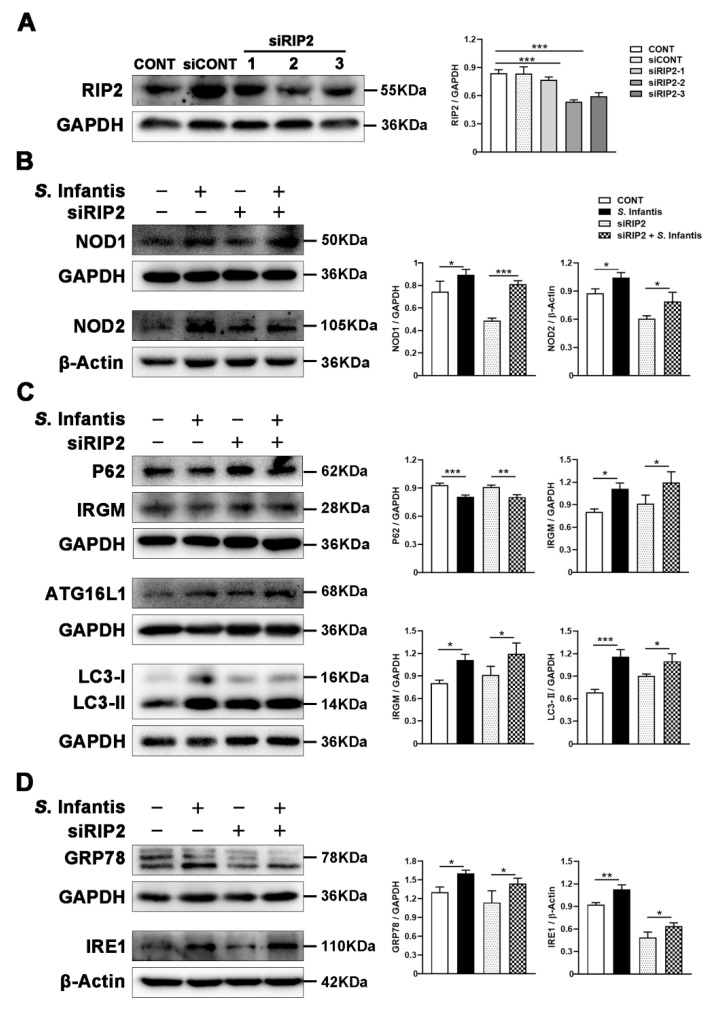
Silencing RIP2 could not block *S*. Infantis-induced autophagy and ER stress in Caco2 cells. (**A**) Western blotting analysis in Caco2 cells treated with siRNA-targeting RIP2. (**B**, **C**, and **D**) Western blotting analysis for NOD1, NOD2, P62, IRGM, ATG16L1, LC3, GRP78, and IRE1 in Caco2 cells. The right panels show the protein quantitation using Image J software. Data are represented as mean ± SEM, *n* = 3; * *p* < 0.05, ** *p* < 0.01, *** *p* < 0.001.

**Figure 7 ijms-23-10395-f007:**
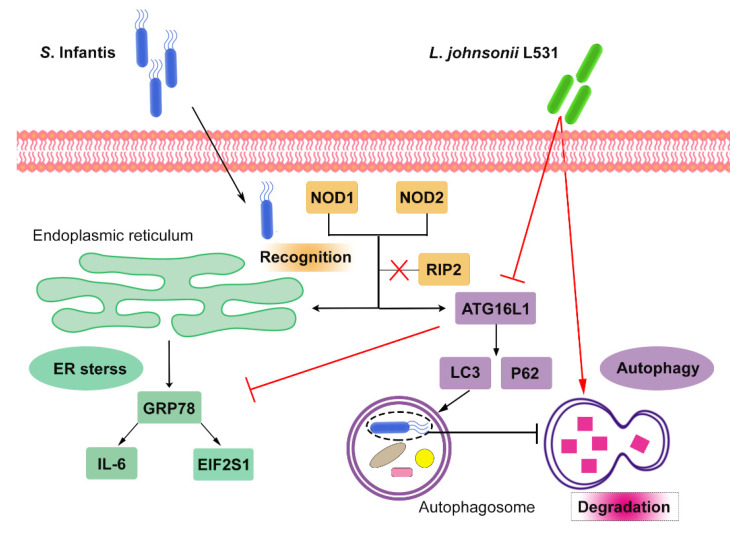
*L. johnsonii* L531 protects the intestine and Caco2 cells from *S*. Infantis infection by alleviating ER stress and promoting autophagy degradation. During *S*. Infantis infection, RIP2 inhibition did not impact the ER stress and autophagy.

**Table 1 ijms-23-10395-t001:** Real-time PCR primers used in porcine samples.

Primer name	Direction	Sequence (5′→3′)	GenBank Accession
GAPDH	F	GATTCCACCCACGGCAAGTTCC	NM_001206359
	R	AGCACCAGCATCACCCCATTTG	
IL-6	F	ATAAGGGAAATGTCGAGGCTGTGC	NM_214399
	R	GGGTGGTGGCTTTGTCTGGATTC	

F = forward; R = reverse.

**Table 2 ijms-23-10395-t002:** Real-time PCR primers used in Caco2 Cells.

Primer name	Direction	Sequence (5′→3′)	GenBank Accession
GAPDH	F	GGAGCGAGATCCCTCCAAAAT	NM_001289746.2
	R	GGCTGTTGTCATACTTCTCATGG	
DDIT3	F	TCTGGCTTGGCTGACTGAGGAG	NM_001195056.1
	R	TTTCCGTTTCCTGGGTCTTCTTTGG	
IL-6	F	GACAGCCACTCACCTCTTCAGAAC	NM_000600.5
	R	GCCTCTTTGCTGCTTTCACACATG	

F = forward; R = reverse.

## Data Availability

Not applicable.

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
