# Peer review of "Lactobacillusjohnsonii L531 Protects against Salmonella Infantis-Induced Intestinal Damage by Regulating the NOD Activation, Endoplasmic Reticulum Stress, and Autophagy"

_ijms, 2022, doi:10.3390/ijms231810395_

Round 1

Reviewer 1 Report

Dear Authors,

your paper is very nice, robust and interesting, The methodologies used are appropriate and the graphics elegantly presented. The argument is not that novel, but the overall work is  well structured.

It would be nice if the auhtors could state at least if the work design and the involvment in animal testing have somehow met the agenda of OneHealth or the 3R

https://www.nc3rs.org.uk/ or https://www.efsa.europa.eu/en/topics/topic/alternatives-animal-testing

Conclusion is poor and should be strenghtened, the authors shoud report and criticall discuss also the limits of their work, propose solutions and give some prespectives. 

There are very few things that need to be revised, as some editing typos, like:

the temperature style is 4 °C, not 4°C

The CFU style has to be harmonized..at line 295 and 296 1.0 × 1011 CFU/mL and 109 CFU/mL

Maybe the references are a bit old and due that are also limited in number, some more recents could be added easily, stenghtening the paper. Just two over 32 are 2021 

Reviewer 2 Report

This work focuses on the potential role of a probiotic organism upon the epithelial effects of a salmonella cultivar. Complementary in vivo and in vitro work utlised to explore this further

SPECIFIC COMMENTS

1. Please ensure that correct formatting is used for bacterial names throughout (including the title)

2. The ASBSTRACT refers to the "establishment" of the two models employed. Although this work used two models of host:bacterial interactions, established is not an appropriate word here

3. line 22: the word "exerted" is also out of place

4. The ABSTRACT refers to beneficial effects of this probiotic in the animal model. The first part of the RESULTS refers further to this. Then we realise that this is prior data. The ABSTRACT should refer to current results arising from this work, not prior results. Presentation of prior data might be relevant in the INTRO or DISCUSSION but not as currently presented. It is unclear from the METHODS whether the current animal studies are independent from the prior work or involve duplication and/or further work on tissues utilised previously

5. line 83. This is an example of an awkward sentence that needs revision/correction. There are numerous other examples of similar sentences that need attention

6. section 2.7 refers to "RIP2 interference". Please consider a different term: inhibition? blockade? 

7. The methods don't justify the dose of bacteria or probiotic utilised, nor do they justify the time periods employed. 

8. Does this probiotic have a treatment effect as well as preventative effect? If translating the current work to real life, one would need to take probiotics prior to any exposure to Salmonella (so not that practical) 

Round 2

Reviewer 2 Report

Thank you for your comments and revisions to date.

1. Regretfully there are still a number of awkward sentences and phrases that require further attention and correction (to further enhance readability). These are in almost every section of the MS. Here are two for example:

- the first line of the ABSTRACT

- page 6: disrupting of the cell membrane integrity. 

2. Abbreviations should be explained in full when first used: NF-kB should be given in full prior to use in the INTRO

3. in the DISCUSSION, the authors refer to the term "discovery". Was this the first demonstration ever of activation of the NOD pathway by this bacterium? If not, the  word discovered should be amended. 

4. The process followed, with further work here building on prior work and available tissue is great. The authors should clearly delineate what is new and what has been already established. These distinctions could be more clearly made

Round 3

Reviewer 2 Report

Thank you for your revisions.

Unfortunately, a number of awkward sentences with inappropriate word usage/structure remain in most parts of the MS

The first sentence of the ABSTRACT contains extra words
